# Higher Prevalence of Dementia but No Change in Total Comfort While Dying among Nursing Home Residents with Dementia between 2010 and 2015: Results from Two Retrospective Epidemiological Studies

**DOI:** 10.3390/ijerph18042160

**Published:** 2021-02-23

**Authors:** Rose Miranda, Tinne Smets, Nele Van Den Noortgate, Luc Deliens, Lieve Van den Block

**Affiliations:** 1End-of-Life Care Research Group, Vrije Universiteit Brussel (VUB) and Ghent University, 1090 Brussels, Belgium; tinne.smets@vub.be (T.S.); luc.deliens@vub.be (L.D.); lieve.van.den.block@vub.be (L.V.d.B.); 2Department of Family Medicine and Chronic Care, Vrije Universiteit Brussel (VUB), 1090 Brussels, Belgium; 3Department of Geriatric Medicine, Ghent University Hospital, 9000 Ghent, Belgium; nele.vandennoortgate@uzgent.be; 4Department of Public Health and Primary Care, Ghent University Hospital, 9000 Ghent, Belgium

**Keywords:** long-term care, care homes, nursing homes, dementia, quality improvement, palliative care

## Abstract

Important policy developments in dementia and palliative care in nursing homes between 2010 and 2015 in Flanders, Belgium might have influenced which people die in nursing homes and how they die. We aimed to examine differences between 2010 and 2015 in the prevalence and characteristics of residents with dementia in nursing homes in Flanders, and their palliative care service use and comfort in the last week of life. We used two retrospective epidemiological studies, including 198 residents in 2010 and 183 in 2015, who died with dementia in representative samples of nursing homes in Flanders. We found a 15%-point increase in dementia prevalence (*p*-value < 0.01), with a total of 11%-point decrease in severe to very severe cognitive impairment (*p* = 0.04). Controlling for residents’ characteristics, in the last week of life, there was an increase in the use of pain assessment (+20%-point; *p* < 0.03) but no change in total comfort. The higher prevalence of dementia in nursing homes with no change in residents’ total comfort while dying emphasizes an urgent need to better support nursing homes in improving their capacities to provide timely and high-quality palliative care services to more residents dying with dementia.

## 1. Introduction

Dementia is a progressive incurable condition, for which a palliative care approach is widely recommended [1]. Palliative care can improve the quality of life of people with dementia by addressing their multi-faceted physical, psychosocial and spiritual care needs for months or years until death [1,2,3]. In Europe, the prevalence of dementia is projected to almost double to about 18.8 million by 2050 [4]. Because people with dementia have prolonged and complex palliative care needs [2], half or more of them eventually live and receive care in nursing homes [5,6]. Yet, the quality of dying and end-of-life care in this setting in many countries, including those with high levels of palliative care development, such as in Belgium, is apparently sub-optimal [7,8]. Especially residents with dementia remain at risk of dying with great discomfort, potentially related to sub-optimal assessment and management of their complex care needs, which highlights an urgent need to identify ways on how to improve the quality of care in nursing homes for this population [2,9,10,11,12].

Over the past decade, there have been important policy developments related to dementia and palliative care in the nursing home sector in many countries, which might have influenced which people die in nursing homes and how they die, and can impact the provision of palliative care for nursing home residents with dementia [8,13]. Yet, there is a lack of high-quality data on the number of residents dying in nursing homes with varying stages of dementia; on the extent to which palliative care services are used; and on how these number of residents dying in nursing homes and their palliative care service use are changing over time. In this study, we will examine differences between 2010 and 2015 in the prevalence and characteristics of residents with dementia in nursing homes in Flanders, the Dutch-speaking part of Belgium where about 60% of the population live [14], as well as differences in their palliative care service use and comfort in the last week of life.

Between 2010 and 2015, new dementia policies in Flanders were oriented towards delaying the institutionalization of older people with dementia by enabling them to stay at home for as long as possible [15,16,17,18,19]. Several initiatives for people with dementia living in communities were also implemented regionwide, e.g., information campaigns and training of primary care professionals and family caregivers [20,21,22,23]. While these developments could potentially result in fewer admissions or shorter stays in nursing homes or more admissions of those with advanced conditions unmanageable at home [24,25], representative data showing these potential changes are lacking. 

Particularly three developments related to palliative care in the nursing home sector in Flanders are historically important. First, supported by the 2002 Belgian Palliative Care Law that recognizes the legal right to palliative care of ‘patients whose life-threatening illness no longer responds to curative treatments’ [26], the Flemish government passed the Decree on Residential Care in 2009 [27]. This decree officially requires Flemish nursing homes to support, sensitize, and train all regular staff regarding palliative care. Second, in 2010, the Flanders’ Federation of Palliative Care launched palliative care guidelines for professional caregivers in nursing homes in developing and implementing palliative care in their facility [28,29], including the comprehensive delivery of physical, psychosocial, and spiritual support [29]. Third, in 2013, the Flemish government introduced a strategy to evaluate the quality of care in nursing homes by having them report on 13 quality indicators [30]. Two of these quality indicators concern palliative care (‘place of death’ and ‘advance care planning’). These indicators are used to systematically monitor the aggregate quality of care in this sector and to identify areas where improvements can be made. Despite these policy developments for nursing homes, no epidemiological study has yet examined the use of palliative care services of residents with dementia and their comfort while dying before and after these developments. Examining this will inform policymakers in Belgium on how to further improve the quality of care at the end of life of nursing home residents with dementia. Results can also be used to inform policies in many countries, especially in Western Europe with similar shifts in health care policies [8,13]. Hence, focusing on Flanders, we sought to answer the following research questions:-Are there differences in the prevalence of dementia in nursing homes between 2010 and 2015?-Are there differences in the socio-demographic and clinical characteristics of nursing home residents with dementia between 2010 and 2015?-Are there differences in palliative care service use and comfort in the last week of the life of nursing home residents with dementia between 2010 and 2015?

## 2. Materials and Methods

### 2.1. Design

We used data from two retrospective epidemiological studies in regionwide representative samples of nursing homes in Flanders, Belgium, namely, the Dying Well with Dementia study focused on residents with dementia (2010) and the Palliative Care for Older People in care and nursing homes in Europe (PACE) study focused on all residents, of whom the presence of dementia was determined (2015) [31,32]. Both studies used similar research methods unless otherwise indicated.

### 2.2. Participating Nursing Homes

To obtain regionwide representative samples of nursing homes, proportional stratified random sampling methods were used. From a national list, the research team randomly sampled Flemish nursing homes, stratified by region (five provinces), bed capacity (up to or more than 90 beds, which is the median number of beds in nursing homes in Flanders), and ownership (public, private/non-profit, private/profit). Previous studies showed that region, bed capacity, and ownership are factors associated with end-of-life care quality in nursing homes [33,34]. If a nursing home refused to participate, another one was randomly selected from the same stratum until the targeted number per stratum was reached [31,32].

### 2.3. Data Collection and Study Population

The administrator/manager in each nursing home was asked to identify all residents who died in the previous three months. Because the 2010 study focused on dementia, the residents who did not have dementia were excluded immediately before data collection. This was done by asking the administrators/managers to further identify residents who met the Katz scale criteria used by the Belgian health insurance system to allocate financial resources: “category Cdementia”, i.e., being completely care-dependent or needing help for bathing, dressing, eating, toileting, continence, and transferring plus being disoriented in time and space OR “disorientation in time and space” (≥3 or “almost daily a problem with disorientation in time and space”) [31,32,35].

In 2010, data were collected on residents who met any of the Katz scale criteria, while in 2015, data were collected on all identified residents. To collect data, both studies used after-death questionnaires distributed to nursing home staff most closely involved in care, general practitioner (GP), and nursing home administrator. Dementia was determined by asking the GP and the nursing home staff if the resident “had dementia” or “was diagnosed with dementia”. We considered a resident to have dementia if the nursing home staff and/or the GP indicated it. A resident did not have dementia when both the nursing home staff and the GP indicated it, or when one of these respondents indicated it, but the other did not return the questionnaire or did not answer the question [31,32].

Response rates for staff, GPs, and administrators were, respectively, 88.4%, 52.9%, and 95.0% in 2010 and 85.1%, 68.3%, and 94.2% in 2015. We excluded residents for whom the nursing home staff did not return the questionnaire. Non-response analysis showed no difference in residents’ characteristics between cases for whom the questionnaire was returned by nursing home staff or not [7,31].

### 2.4. Measurements and Outcome Measures

#### 2.4.1. Residents’ Characteristics

Using validated instruments, the nursing home staff reported residents’ cognitive and functional impairment one month before death. Cognitive Performance Scale (CPS) uses five variables from the Minimum Data Set to group residents into six hierarchical cognitive performance categories, e.g., CPS scores 5–6 indicate severe and very severe impairment [36]. Global Deterioration Scale (GDS) is divided into seven stages, of which stage 7 indicates that a resident lost all verbal abilities, was incontinent/required assistance with eating and toileting, and lost basic psychomotor skills [37]. Hence, to determine whether a resident had GDS stage 7, the nursing home staff were asked whether the resident fit all the criteria of GDS stage 7 (yes/no). To compare with earlier studies [7,31], we determined the severity of dementia using CPS and GDS (CPS scores ≥ 5 and GDS stage = 7 had advanced dementia, while the rest had non-advanced dementia). The nursing home staff also reported the Bedford Alzheimer Nursing Severity scale (BANS-S), with total scores ranging from 7 (no impairment) to 28 (complete impairment) [38]. They also reported whether any clinical complication occurred in the last month of life, e.g., pneumonia or intake problems. The GPs reported co-existing conditions, e.g., cancer or cardiovascular disease. Nursing home administrators reported residents’ age at time of death, gender, length of stay in nursing homes, place of death, and whether the residents stayed in an open or secured unit at the time of death and in a dementia care unit or not. These residents’ characteristics could influence the palliative care service use and comfort at the of life of people with dementia [39,40,41,42].

#### 2.4.2. Palliative Care Services Used

The nursing home staff reported data on palliative care service use, including (1) whether a palliative care record was initiated for residents and the days before death when this occurred; (2) whether a resident received palliative care at any time, including whether this palliative care was provided by a GP and whether the following persons/initiatives were involved in providing this care: coordinating and advisory physician, palliative care reference nurse, palliative care task group, specialist palliative home care team, or none of them. Since 2009, nursing homes in Flanders were officially required to establish a functional relationship with general practitioners (GPs) responsible for providing medical care and developing palliative care strategies for residents and coordinating and advisory physicians responsible for coordinating with GPs to review palliative care strategies and give advice and training to staff [27,43]. Further, the nursing homes must have a palliative care reference nurse responsible for establishing a supportive palliative care culture and awareness within the nursing home, training personnel regarding palliative care, and supporting and coordinating palliative care delivery, and a palliative care task group comprising of all palliative caregivers. For complex palliative situations, palliative home care teams can either call or visit nursing homes to provide advice or support [27,43]. The nursing home staff also reported whether the residents received services related to medical or nursing treatments/procedures in the last week of life, psychosocial interventions in the last month, and spiritual and/or pastoral care before death.

#### 2.4.3. Comfort in the Last Week of Life

They also assessed comfort in the last week of life using the Comfort Assessment in Dying-End-of-Life in Dementia (CAD-EOLD) scale. CAD-EOLD is a validated 14-item scale comprising discomfort, pain, restlessness, shortness of breath, choking, gurgling, difficulty swallowing, fear, anxiety, crying, moaning, serenity, peace, and calm. Individual item scores range from 0–3, while total scores range from 14 to 42, with higher scores representing better comfort [44,45].

### 2.5. Data Analyses

The 2010 and 2015 databases were merged by R.M. and two palliative care researchers. The prevalence of dementia between 2010 and 2015 was compared using *χ*^2^-test. Subsequent analyses were performed in IBM SPSS statistics version 26 (©IBM Corporation; Armonk, NY, USA) using generalized linear mixed model to account for clustering of data within nursing homes. We compared residents’ characteristics and their palliative care service use and comfort scores between 2010 and 2015. We adjusted all analyses related to palliative care service use and comfort for resident characteristics while taking correlations between these resident characteristics into account. Using Benjamini-Hochberg procedure to decrease the false discovery rate, we adjusted the analyses related to comfort for multiple testing. Hypothesis testing was two-sided. Statistical significance was set at *p* < 0.05.

## 3. Results

### 3.1. Prevalence of Dementia

The prevalence of dementia significantly increased from 43% in 2010 (205 of 477 residents) to 58% in 2015 (199 of 342 residents) (+15%-point; *p*-value < 0.01; Figure 1). Of the residents with dementia, we excluded 7 residents in 2010 and 16 in 2015, as the nursing home staff did not return the questionnaires, leaving 198 and 183 residents for further analyses. In the large majority of nursing homes in both years, the number of residents in each nursing home ranged between 1 and 8. In 2010, two nursing homes had 11 and 14 residents, while in 2015, one nursing home had 9 residents.

### 3.2. Characteristics of Residents with Dementia

Between 2010 and 2015, residents’ characteristics did not change, except for scores on the Cognitive Performance Scale. One month before death, the proportion of residents with dementia with severe to very severe cognitive impairment (CPS scores 5-6) had a total of 11%-point decrease (*p* = 0.04; Table 1), while the proportion of residents with GDS stage 7 had a total of 14%-point increase (*p* = 0.04). The residents were about 86 years of age at the time of death, were predominantly women, and had BANS-S scores of 20.9 in 2010 and 20.3 in 2015. Of the residents in 2010 and 2015, respectively, 49% and 52% had advanced dementia, while 95% and 92% experienced any clinical complication a month before death. The most common co-existing conditions were cardiovascular diseases (29% in 2010 and 28% in 2015), followed by cancer and respiratory conditions. The median length of stay in nursing homes was 893 days in 2010 and 688 days in 2015. In 2010 and 2015, respectively, nursing home was the most common place of death (90% and 86%), while 9% and 14% died in hospitals.

### 3.3. Palliative Care Service Use among Residents with Dementia

In the multivariable analyses controlled for residents’ characteristics, in 2010 and 2015, respectively, a palliative care record was initiated for 62% and 72% of residents (*p* = 0.17), of which 51% and 60% occurred within 14 days before death (*p* = 0.63; Table 2). According to nursing home staff, 83% in 2010 and 82% in 2015 of residents received palliative care. For 17% (2010) and 20% (2015) of these people who received palliative care, no coordinating and advisory physician, palliative care reference nurse, palliative care task group, and palliative home care teams were involved (*p* = 0.83).

In the last week of life, there was a significant increase in the percentages of residents for whom pain assessment was conducted (from 63% in 2010 to 83% in 2015; *p* = 0.03). In the last month of life, 37% (2010) and 47% (2015) of residents did not receive any psychosocial intervention (*p* = 0.78). In 2010 and 2015, respectively, shortly before death, 48% and 57% of residents received spiritual care, meaning that 52% and 43% did not receive it (*p* = 0.11).

### 3.4. Comfort in the Last Week of Life

In multivariable analyses controlled for residents’ characteristics, a week before death, there was a 0.2-point increase in the comfort scores related to moaning (*p* = 0.03) (Table 3). However, this statistically significant increase in comfort scores disappeared after adjusting for multiple testing (*p* = 0.45). The estimated marginal means for the total comfort scores did not change between 2010 (30.0; 95% CI = 29.2–30.8) and 2015 (30.8; 29.2–30.9; *p* = 0.87).

## 4. Discussion

Our study showed that between 2010 and 2015 in nursing homes in Flanders, Belgium, there was a 15%-point increase in the prevalence of dementia. Almost all residents’ characteristics did not change, except for the level of cognitive impairment in the last month of life, with a total of 11%-point decrease in residents with severe and very severe cognitive impairment, and the level of cognitive and functional impairment, with a total of the 14%-point increase in residents who lost all verbal abilities, was incontinent/required assistance with eating and toileting and lost basic psychomotor skills. The percentages of residents with advanced dementia were 49% in 2010 and 52% in 2015. Pain assessment in the last week of life was performed proportionally more often for residents in 2015 than in 2010. However, in both years, between 37% and 52% of residents neither received psychosocial intervention in the last month of life nor spiritual care shortly before death. In the last week of life, we found no change in residents’ total comfort in the last week of life. 

This is the first time that two retrospective epidemiological studies are used to investigate changes over time for residents with dementia in the context of important developments in the landscape of dementia and palliative care policies and initiatives in nursing homes. Retrospective data collection is a feasible method for population-based epidemiological end-of-life studies, as it limits potential bias in prospective sampling, e.g., underrepresentation of people who live longer than the follow-up period [7]. Although these are separate studies, both utilized similar study designs, aiming to reach representative samples, and all variables of interest were measured in the same way. Finally, while the measurement of palliative care services is limited to services measured in both studies, these services comprise important components of palliative care in dementia, e.g., comprehensive delivery of physical, psychosocial, or spiritual support [1]. However, this study also has limitations. As these are two separate studies, and the study in 2010 primarily focused on dying nursing home residents with dementia, the variables that could be explored and compared between the years were limited, especially on nursing home characteristics, that might influence palliative care service use or comfort. While accounting for the clustering of data within nursing homes in the analyses could partly limit this limitation of our data, our inability to control for unmeasured variables that could influence palliative care service use or comfort remains a clear limitation of our study. Because data were collected after death, there might be some recall bias [7]. Further, only 2010 Dying Well with Dementia study used the Katz-scale criteria to exclude residents without dementia before data collection [35]. Nevertheless, such residents without dementia would have also been identified by the nursing home staff and/or the GPs in the PACE study, as they were involved closely in resident care [46,47]. For 19 residents in 2010 and 51 residents in 2015, we could not determine the presence or absence of dementia, which may influence the prevalence of dementia. In certain variables, such as the CAD-EOLD, we have a relatively large proportion of missing values (>5%), which we have reported in detail in the footnotes of Table 1, Table 2 and Table 3. Finally, given the cross-sectional nature of the study, it is not possible to identify explanations for the findings within our study. For instance, we could not explore whether the extent of residents’ palliative care service use relates to the identified lack of change in their total comfort in the last week of life (i.e., temporal relationship).

Our study clearly showed that between 2010 and 2015, there is a substantially higher prevalence of nursing home residents with dementia with very minimal change in their clinical and socio-demographic characteristics. Over this relatively short period, almost an additional 15% of the residents die with dementia. Perhaps, this is because such increase in the prevalence of dementia also occurred in the home setting, as the 2016 estimates in Flanders suggest that there were 15,855 more people with dementia in 2015 than in 2010 [48], which is congruent with the current trends in dementia prevalence in other countries in Europe [4]. At the end of life, people with dementia also have complex care needs that could complicate primary care delivery and could thus become unmanageable at home [11,12]. Hence, more people with dementia living at home may have been transferred eventually to nursing homes [24,25]. Further, over the years, nursing home residents with dementia apparently remain to have almost similar clinical and demographic characteristics, which suggests that their complex and prolonged care needs at the end of life persist over the years [11,12]. We found that among residents in 2010 and 2015, about half had advanced dementia, more than 90% developed any clinical complication in the last month of life, and the majority stayed in nursing homes for about two years. While we found a somewhat lower percentage of residents who died with severe cognitive impairment (i.e., CPS scores 5-6) in 2015 than in 2010, the percentage of residents who lost all verbal abilities, was incontinent/required assistance with eating and toileting, and lost basic psychomotor skills (i.e., GDS stage 7) increased over the years. These findings might explain the slightly higher but non-statistically significant difference in the proportion of residents with advanced dementia in 2015 than in 2010. The identified lower proportion of residents with severe cognitive impairment based on CPS scores suggests that these residents died from other diseases that do not result in cognitive impairment. Comorbidities, which often occur alongside old age and dementia, present additional challenges for nursing home staff and healthcare service delivery to residents living and dying with dementia [49].

In addition, our study showed that in the last week of the life of residents with dementia between 2010 and 2015, there was an increase in their use of medical/nursing procedures, in particular pain assessment. This is encouraging, as pain is highly prevalent among older people with dementia [50]. However, the use of other medical/nursing procedures, psychosocial interventions, and spiritual care at the end-of-life seemed to lag behind. For instance, the residents’ use of assistance with eating and drinking did not change over time, which needs urgent attention, as intake problems are common in advanced dementia [11,12]. Further, there was still a substantial proportion of residents with dementia, who neither received psychosocial interventions nor spiritual care at the end of life. These findings underscore the persistent lack of attention given to the comprehensive care encompassing physical, psychosocial, and spiritual support, which are paramount to improving residents’ overall comfort at the end-of-life [1].

Promoting comfort for nursing home residents with dementia is a key policy goal of care in many countries and a palliative care approach has been widely advocated to improve comfort in this population [1,51,52,53,54,55]. However, providing high-quality and comprehensive palliative care to and improving comfort in nursing home residents with dementia is a highly demanding and complex work for care professionals [40,41]. Our identified increase in the prevalence of nursing home residents with dementia and the minimal change in the complexity of their care needs at the end-of-life highlight the increasing complexity of the challenges faced by the nursing home sector. This evolution is likely to continue in the future, as the prevalence of dementia in Flanders has been projected to almost double by 2060 [48]. Such evolution might also be comparable with evolution in other countries that implemented similar dementia and palliative care policies and initiatives and have similarly increasing dementia prevalence [4,8,13]. Further, we found that despite an encouraging improvement in the use of pain assessment of residents with dementia, there remains a lack of change in their total comfort in the last week of life. In order to better support nursing home staff to maintain the high quality of care in nursing homes and to improve comfort at the end-of-life of a growing number of residents with dementia [9,48], there is an urgent need for continued and stronger public health investments and a more comprehensive palliative care approach in this sector [1]. The timely and consistent implementation of comprehensive palliative care in dementia approach requires a strong national and regional policy commitment and the incorporation of this approach in the attitudes and skills of nursing home staff [56,57]. Because there is still no known effective palliative care program for nursing home residents with dementia [58], future research should continue developing and evaluating palliative care programs that could improve comfort at the end-of-life in this population. Strategies on how to develop, implement, and evaluate complex palliative care interventions in nursing homes and the factors that need to be addressed in doing so have been published [59,60,61].

## 5. Conclusions

Our study suggests that between 2010 and 2015, there was a higher prevalence of residents with dementia in nursing homes in Flanders, Belgium who persistently have complex care needs at the end-of-life. Further, despite an encouraging improvement in the use of pain assessment of residents with dementia, there remains a lack of change in their total comfort in the last week of life. These findings highlight the increasing complexity of challenges faced by the nursing home sector, which underscores an urgent need to better support nursing homes in improving their capacities to provide timely, high-quality, and comprehensive palliative care to a growing number of nursing home residents living and dying with dementia.

## Figures and Tables

**Figure 1 ijerph-18-02160-f001:**
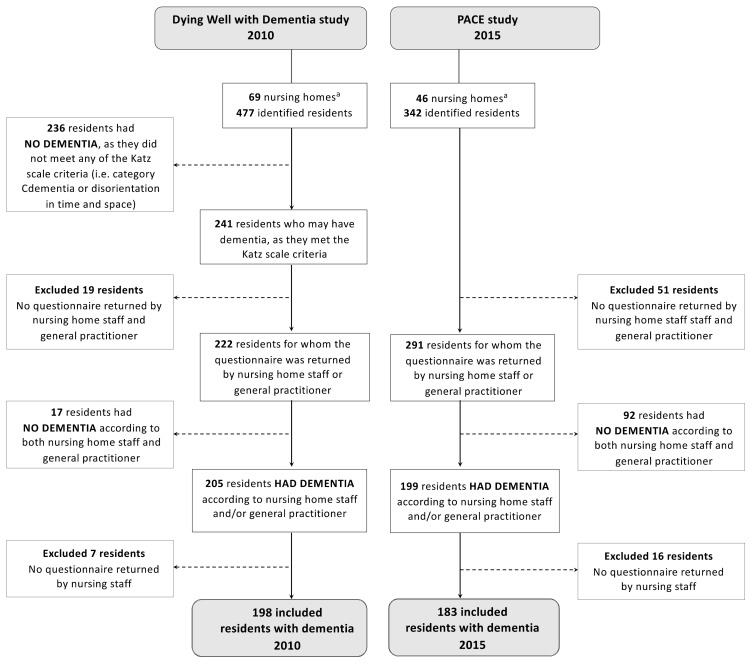
Overview of the identification of residents with dementia in 2010 and 2015. ^a^ Because we excluded a number of residents from the total sample, the final number of nursing homes were 64 in 2010 and 43 in 2015. In 2010, 205 residents had dementia (numerator) of the 477 identified residents. In 2015, 199 residents had dementia (numerator) of the 342 identified residents (denominator).

**Table 1 ijerph-18-02160-t001:** Comparing residents’ characteristics between 2010 and 2015.

Residents’ Characteristics	2010 (N = 198)	2015 (N = 183)	Change	Difference between the Years *p*-Values †
Socio-demographic characteristics				
Age at time of death, average in years (SD)	86.7 (7)	86.9 (7.3)	+0.2	0.73
Gender, female *n* (%)	115 (61)	114 (65)	+4	0.43
Clinical characteristics				
Cognitive performance scale (CPS), *n* (%)				0.04
- Intact, borderline intact, mild impairment (score 0-1-2)	8 (4)	21 (13)	+9	
- Moderate impairment (score 3)	27 (14)	20 (12)	−2	
- Moderately severe impairment (score 4)	9 (5)	15 (9)	+4	
- Severe impairment (score 5)	61 (33)	52 (31)	−2	
- Very severe impairment (score 6)	82 (44)	58 (35)	−9	
Global deterioration scale (GDS) stage 7, *n* (%)	105 (58)	123 (72)	+14	0.04
Bedford Alzheimer Nursing Severity scale (BANS-S) 1 month before death, mean (SD)	20.9 (3.9)	20.3 (4.3)	−0.6	0.19
Severity of dementia, *n* (%)				0.55
- Non-advanced dementia	95 (51)	75 (48)	−3	
- Advanced dementia	92 (49)	82 (52)	+3	
Occurrence of clinical complications in last month of life, *n* (%)	179 (95)	168 (92)	−3	0.31
Co-existing conditions				
- Cancer	12 (11)	19 (14)	+3	0.43
- Cardiovascular	32 (29)	37 (28)	−1	0.89
- Respiratory	15 (14)	14 (11)	−3	0.51
- Neurological (not dementia)	17 (15)	10 (8)	−7	0.08
- Urogenital	9 (8)	12 (9)	+1	0.82
- Other	18 (16)	22 (17)	+1	0.99
Length of stay in nursing home in days, median days (IQR)	893 (448–1694)	688 (283–1678)	−205	0.28
Place of death, *n* (%)				0.20
- Nursing homes	171 (90)	152 (86)	−4	
- Hospital	17 (9)	24 (14)	+5	
- Other ‡	2 (1)	0 (0)	−1	
Type of unit at time of death, *n* (%)				0.71
- Open unit	91 (48)	88 (50)	+2	
- Secured unit	98 (51)	88 (50)	−1	
Dementia care unit, yes, *n* (%)	99 (53)	93 (53)	0	0.95

SD = standard deviations; IQR = Interquartile range; GLMM = generalized linear mixed model analyses. † Calculated using GLMM to account for correlation of data within nursing homes; ‘other’ categories not included in calculation of *p*-values. Analyses showed correlation of CPS with GDS, BANS-S and severity of dementia, and this might be because they similarly cover residents’ cognitive and functional status. The type of unit at the time of death was correlated with dementia care unit, and this might be because one nursing home can have both types of unit. Further analyses will be adjusted for CPS, dementia care unit, and the rest of the residents’ characteristics. ‡ Examples of places of death other than nursing homes or hospitals include facility hospice/palliative care unit. Missing values, *n*: age, 2010 = 12; 2015 = 8 | gender, 2010 = 11; 2015 = 9 | severity of dementia, 2010 = 11; 2015 = 26 | CPS, 2010 = 11; 2015 = 17 | GDS, 2010 = 16; 2015 = 12 | BANS-S, 2010 = 4; 2015 = 2 | clinical complications, 2010 = 10; 2015 = 15 | all co-existing conditions except other, 2010 = 87; 2015 = 51 | other co-existing conditions, 2010 = 88; 2015 = 51 | length of stay in nursing homes, 2010 = 13; 2015 = 10 | place of death, 2010 = 10; 2015 = 7 | type of unit, 2010 = 9; 2015 = 7 | dementia care unit, 2010 = 10; 2015 = 8.

**Table 2 ijerph-18-02160-t002:** Comparing palliative care service use between 2010 and 2015.

Palliative Care Service Use	2010 (N = 198)	2015 (N = 183)	%-Point	Difference between Years (*p*-Values ‡)	
	*n* (%)	*n* (%)	Change †	Crude	Adjusted
Residents who had a palliative care record	121 (62)	97 (72)	+10	0.10	0.17
Time before death when the palliative care record initiated					
- <14 days	51 (51)	38 (60)	+9	0.10	0.63
- 15 to 90 days	32 (32)	22 (35)	+3		
- >90 days	18 (18)	3 (5)	−13		
Residents who received palliative care at any time according to nursing home staff	162 (83)	145 (82)	−1	0.69	0.21
Palliative care was provided by GP	136 (84)	123 (86)	+2	0.84	0.89
Other person/initiatives involved in providing the palliative care					
- Coordinating and advisory physician	44 (27)	35 (23)	−4	0.44	0.11
- Palliative care reference nurse	110 (66)	94 (62)	−4	0.64	0.35
- Palliative care task group within the nursing home	81 (49)	64 (42)	−7	0.35	0.34
- Palliative home care teams (external)	16 (10)	8 (5)	−5	0.24	0.30
- No one from this list was involved	28 (17)	30 (20)	+3	0.55	0.83
Residents who received medical or nursing treatments/procedures during the last week of life					
Mouthcare	159 (80)	152 (88)	+8	0.055	0.54
Pain assessment	124 (63)	143 (83)	+20	0.001	0.03
Prevention of pressure ulcers	162 (82)	151 (87)	+5	0.15	0.72
Wound care	45 (23)	48 (28)	+5	0.27	0.97
Assistance with eating/drinking	142 (72)	141 (82)	+10	0.04	0.37
Residents who received psychosocial interventions in the last month of life					
Adjustments of environmental factors ¶	19 (10)	28 (16)	+6	0.10	0.18
Activity programmes	25 (13)	16 (9)	−4	0.33	0.85
Music therapy	48 (24)	28 (16)	−8	0.17	0.24
Behavioural therapy	0 (0)	1 (1)	+1	0.87	0.78
Experiential approaches #	52 (26)	47 (28)	+2	0.90	0.32
No psychosocial interventions received	74 (37)	81 (47)	+10	0.15	0.78
Residents who received spiritual and/or pastoral care shortly before death					
Spiritual care provider/Pastoral worker	98 (48)	72 (57)	+9	0.10	0.11

GLMM = generalized linear mixed model; GP = general practitioners; pp = percentage point. Crude model is the unadjusted model. Adjusted model is adjusted for all residents’ characteristics, except for GDS, BANS-S, severity of dementia, and type of unit at the time of death to avoid multi-collinearity. † %-point = percentage point. %-point difference was calculated between 2010 and 2015. ‡ Calculated using GLMM analyses to account for correlation of data within nursing homes while accounting for differences in resident characteristics; ‘other’ categories not included in the calculation of *p*-values. ¶ Example of adjustments of environmental factors includes a modified environment for walking around safely. # Examples of experiential approaches include multisensory environment, validation therapy. Missing values, *n*: palliative care record, 2010 = 2; 2015 = 49 | receipt of palliative care, 2010 = 4; 2015 = 6 | palliative care provided by GP, 2010 = 5; 2015 = 9 | time before death when palliative care record was started, 2010 = 22; 2015 = 34 | all physical care, 2015 = 10 | all psychosocial care, 2015 = 12 | spiritual care, 2010 = 10; 2015 = 15.

**Table 3 ijerph-18-02160-t003:** Comparing comfort in the last week of life between 2010 and 2015.

COMFORT IN THE LAST WEEK OF LIFE	2010 (N = 198)	2015 (N = 183)	Score-Point	Difference between Years (*p*-Values ‡)	
CAD-EOLD individual items	CAD-EOLD scores 0 (worst) to 3 (best)	CAD-EOLD scores 0 (worst) to 3 (best)	Change †	Crude	Adjusted
- Discomfort	2.1 (2.0–2.2)	2.1 (2.0–2.2)	–	0.46	0.88
- Pain	2.0 (1.9–2.1)	2.2 (2.1–2.3)	+0.2	0.03	0.62
- Restlessness	2.1 (2.0–2.2)	2.1 (2.0–2.2)	–	0.72	0.39
- Shortness of breath	2.2 (2.1–2.3)	2.4 (2.3–2.5)	+0.2	0.03	0.14
- Choking	2.1 (2.0–2.2)	2.1 (2.0–2.2)	–	0.77	0.75
- Gurgling	2.3 (2.2–2.4)	2.5 (2.3–2.6)	+0.2	0.13	0.83
- Difficulty swallowing	1.9 (1.8–2.0)	1.9 (1.8–2.0)	–	0.61	0.84
- Fear	2.0 (1.9–2.2)	2.2 (2.1–2.3)	+0.2	0.04	0.45
- Anxiety	2.1 (2.1–2.3)	2.2 (2.1–2.3)	+0.1	0.32	0.88
- Crying	2.7 (2.6–2.8)	2.7 (2.6–2.8)	–	0.49	0.89
- Moaning	2.3 (2.3–2.4)	2.5 (2.4–2.6)	+0.2	0.02	0.03
- Serenity	2.0 (1.9–2.2)	2.1 (2.0–2.2)	+0.1	0.69	0.07
- Peace	2.0 (1.9–2.1)	2.0 (1.9–2.2)	–	0.63	0.24
- Calm	2.0 (1.9–2.1)	2.0 (1.9–2.1)	–	0.33	0.31
Total score ^¶^, estimated marginal means (95% CI)	30.0 (29.2–30.8)	30.8 (29.2–30.9)	+0.8	0.22	0.87

CAD-EOLD = Comfort Assessment in Dying—End of Life in Dementia; CI = confidence intervals. Crude model is the unadjusted model. Adjusted model is adjusted for all residents’ characteristics, except for GDS, BANS-S, severity of dementia and type of unit at the time of death to avoid multi-collinearity. † Score point change was calculated between 2010 and 2015. ‡ Calculated using GLMM analyses to account for correlation of data within NHs while accounting for differences in resident characteristics. ¶ Total scores are averages per whole scale multiplied by total number of items (i.e., 14). Cases with missing values on more than 25% of items per scale were excluded from total score calculation; scores range from 14 to 42; higher scores indicate better comfort when dying. Missing values, *n*: discomfort, 2010 = 19; 2015 = 12 | pain, 2010 = 9; 2015 = 10 | restlessness, 2010 = 15; 2015 = 10 | shortness of breath, 2010 = 12; 2015 = 10 | choking, 2010 = 16; 2015 = 9 | gurgling, 2010 = 18; 2015 = 11 | difficulty swallowing, 2010 = 11; 2015 = 11 | fear, 2010 = 13; 2015 = 10 | anxiety, 2010 = 14; 2015 = 10 | crying, 2010 = 17; 2015 = 10 | moaning, 2010 = 16; 2015 = 9 | serenity, 2010 = 16; 2015 = 12 | peace, 2010 = 18; 2015 = 12 | calm, 2010 = 19; 2015 = 12 | total score, 2010 = 16; 2015 = 10.

## Data Availability

The data that support the findings of this study are available from the corresponding author upon reasonable request.

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
