# Peer review of "Higher Prevalence of Dementia but No Change in Total Comfort While Dying among Nursing Home Residents with Dementia between 2010 and 2015: Results from Two Retrospective Epidemiological Studies"

_ijerph, 2021, doi:10.3390/ijerph18042160_

Round 1
Reviewer 1 Report
Thanks for the opportunity to review the manuscript. Here are my comments.
- I am not sure the prevalence of dementia was calculated correctly. “The prevalence of dementia significantly increased from 45% in 2010 (205 of 458 residents) to 68% in 2015 (199 of 291 residents) (+23%-point; P-value<0.001; Figure 1).” If I understand Figure 1 correctly, 205 residents who had dementia died in the study of 2010. However, to calculate prevalence of dementia, the numerator should include all residents with dementia regardless of death. I cannot tell how to get the number of 458. For the survey in 2015, I think there were 342 residents which should be the denominator.
- There are 198 participants from 69 nursing homes in 2010 and 183 participants from 46 nursing homes in 2015. What was the range for the number of participants in each nursing home? It seems some nursing homes may have only a few participants.
- There is no information about the characteristics of nursing homes. It is important to compare the changes in nursing home characteristics between 2010 and 2015. For example, the authors found that there was a significant increase in the percentages of residents for whom pain assessment was conducted. This increase in pain assessment may be possibly due to the increase in the number of nursing staff.
- Without a cohort study to tract the changes in comfort scores at the end of life, it is hard to reach the conclusion that there is no improvement in total comfort while dying between 2010 and 2015. There were policies implemented to delay the institutionalization of older people with dementia. It is possible that residents with dementia in 2015 had more complex care needs and lower baseline comfort scores than those in 2010 at the end of life. Even though the palliative care improved the comfort score in 2015, it is still possible that there was no improvement between 2010 and 2015 in the last week of life.
- More detailed information is needed in the data analysis section. How missing data was handled? What variables were controlled in the models? What were the outcome variables of interest? It seems there were binary outcomes such as whether pain assessment was conducted. It is not appropriate to use the generalized linear models to conduct data analysis when the outcome is not a continuous variable.
- “One month before death, the proportion of residents with dementia with severe to very severe cognitive impairment (CPS scores 5-6) had a total of 11%-point decrease.” However, the percentage of residents with advanced dementia increased between 2010 and 2015 (49% in 2010 and 52% in 2015). It seems the above information is not consistent.
Author Response
Please see the attachment for our detailed response to the reviewer's comment. We would like to thank the reviewer for providing us critical feedback and comments, which we use to further improve our manuscript. We hope to have satisfactorily addressed all the concerns and comments. The texts that were added or revised in the manuscript are set in bold.

Reviewer 2 Report
Thank you for allowing me to review this paper, very interesting.
line 39: the second part of the sentence is unclear
line 83: Write the PACE acronym in full as this is the first time it has been used
line 90-91 : indicate why the cutoff of 90 beds was chosen
line 93: indicate as bibliography the papers justifying the stratification
line 100: insert reference for Katz scale
line 108: Write the GP acronym in full as this is the first time it has been used.
lines 110-111: sentence unclear
Lines 110-111: sentence unclear
Table 1: put the brackets of the first line on the same line
Has the economic impact been investigated and compared?
I would suggest indicating in the introduction that the problem of palliative care in older people is worldwide or only in Belgium.
I would suggest to indicate and underline in the conclusions what are the effects of the results of this study on nursing care, what benefits will be gained by patients and nurses.
I would also suggest to comment on non-significant data that might be relevant for nursing care or might be interesting for further studies.
In addition, I think it would be useful to make some reflections and proposals about ethics.
Author Response
Please see the attachment for our detailed response to the reviewer's comments. We thank the reviewer for reviewing our work and providing suggestions to better improve our article. We hope to have satisfactorily addressed all the concerns and comments.The texts that were added or revised in the manuscript are set in bold.

Round 2
Reviewer 1 Report
Thanks for the detailed response to my previous comments. I believe the manuscript has been improved. However, I still have some comments.
- Since a number of residents (19 in 2010 and 51 in 2015) could not be determined the presence of dementia, an additional limitation may be added in the discussion section. For example, 51/342 = 14.9%. It is a large proportion (nearly 15% in 2015) without dementia information, which may influence the prevalence of dementia.
- Regarding the missing data, listing the number of missing values for variables is good but not enough. The authors compared the differences in missing data between 2010 and 2015. They may also need to compare the differences between missing and non-missing data and consider imputation for missing data. If not possible, a limiting may be added since there is a large proportion of missing (>5%) in certain variables.
- I am satisfied with the way the authors addressed the comment related to CPS and GDS.
- The generalized linear mixed models (GLMM) were used to account for the clustering of residents within the nursing homes. I have several questions. First, does the GLMM have a requirement for the minimum number of residents within one nursing home? In this study, some nursing homes only had one resident (the number of residents in each nursing home ranged between 1 and 8). Second, I assume the nursing home was treated as a random effect. Can this account for the time-varying nursing home characteristics? For example, there might be changes in nursing home staff during 2010-2015. Third, it seems only variables that were different between 2010 and 2015 were controlled in the model, such as differences in CPS scores. Some variables may help explain the variations in the outcomes of interest, although there were no statistical differences between 2010 and 2015. Should these variables be controlled in the model?
- It seems there is a multiple comparisons problem. For example, in table 3, there are 15 outcomes that may involve multiple simultaneous statistical tests. The authors may consider Bonferroni correction or other methods to compensate for multiple comparisons.
